# Improving training on hepatitis B research in Nigeria: Findings from an innovation bootcamp to strengthen capacity

Maria A. Afadapa[1,2*¤a¤b◔], Olufunto A. Olusanya[1◔], Peter Kalulu[1‡],
Nkiruka Obodoechina[1‡], Sonam J. Shah[3‡], Temitope Ojo[1], Abideen Salako[4],
Joseph Ogbeh[4], Folahanmi T. Akinsolu[4], Ucheoma Nwaozuru[5], Adesola Z. Musa[4],
Titilola Gbaja-Biamila[4], Olaitan H. Olayiwola[6‡], Idris A. Oladosu[6‡], Suzanne Day[3],
Peyton Thompson[3], Kristie Foley[5], Oluwaseun Falade-Nwulia[7], Dan Wu[3],
Olufunmilayo Lesi[8], Collins O. Airhihenbuwa[9], Joseph D. Tucker[3,10], Oliver Ezechi[4],
Juliet Iwelunmor[1]

1 Department of Medicine, Washington University in Saint Louis, St. Louis, Missouri, United States of America, 2 Department of Pediatric Infectious Diseases, Lagos State University Teaching Hospital, Lagos, Nigeria, 3 Department of Pediatric Infectious Diseases, University of North Carolina at Chapel Hill School of Medicine, Chapel Hill, North Carolina, United States of America, 4 Department of Clinical Research, Nigeria Institute of Medical Research, Lagos, Nigeria, 5 Department of Implementation Science, Wake Forest University School of Medicine, Winston-Salem, North Carolina, United States of America, 6 Department of Medicine, College of Medicine, University of Ibadan, Oyo, Nigeria, 7 Department of Infectious Diseases, Johns Hopkins University School of Medicine, Baltimore, Maryland, United States of America, 8 Department of HIV, Hepatitis & STIs, World Health Organization Regional Office for Africa, Geneva, Switzerland, 9 Department of Health Policy and Behavioral Science, Georgia State University School of Public Health, Atlanta, Georgia, United States of America, 10 Department of Clinical Research, Faculty of Infectious and Tropical Diseases, London School of Hygiene and Tropical Medicine, London, United Kingdom

◔ These authors contributed equally to this work.
‡ These authors are joint senior authors on this work.
¤a Current address: Department of Medicine, Washington University in Saint Louis, St. Louis, Missouri, United States of America
¤b Current address: Department of Public Health Sciences, The Brown School of Social Sciences and Public Health, Washington University in Saint Louis, St. Louis, Missouri, United States of America
* a.mariaaugustine@wustl.edu

## Abstract

We organized an Innovation Bootcamp for young people in Nigeria to develop strategies promoting the uptake of the hepatitis B birth dose vaccine among newborns, which remains low, with an estimated coverage of 52%. This event was a collaborative, cross-disciplinary capacity-building platform to generate creative solutions addressing barriers to Hepatitis B vaccine uptake. The purpose of this study was to describe the bootcamp activities that address this gap and evaluate the impact of an Innovation Bootcamp designed to build research capacity among young Nigerians. The bootcamp was informed by Participatory Action Research focused at engaging young people as co-researchers to investigate and address issues affecting their respective communities using the PEN-3 cultural model. Qualitative data from the community needs assessment were analyzed using a thematic analysis framework

**Data availability statement:** All relevant data are within the paper and its Supporting Information files.

**Funding:** This work was supported by the National Institutes of Health (1u54ca284110-01 to JDT; 1u54ca284110-01 to OE; 1u54ca284110-01 to JI). The funder had no role in study design, data collection and analysis, decision to publish, or preparation of the manuscript.

**Competing interests:** The authors have declared that no competing interests exist.

to identify and synthesize emerging themes. Socio-demographic characteristics of the participants were collected, and a pre- and post-survey was administered. The participants' knowledge of hepatitis B and research skills were compared using the Wilcoxon test. Our results included five teams composed of fifteen participants with a mean age of 25.5 years, originating from the Southern regions of Nigeria. The post-survey showed significant improvements in participants' knowledge and research skills, with knowledge increasing by 21.6% (mean score: 39.7 to 48.3, $p = 0.010$) and research skills by 36.4% (mean score: 56.1 to 76.5, $p < 0.001$). Each team co-designed implementation strategies, including referral pathways from traditional birth attendants to formal health centers, comprehensive training workshops, and trusted community leaders as vaccination ambassadors. In conclusion, the bootcamp demonstrated its effectiveness in strengthening capacity and increasing knowledge (although in a small sample size), which contributed to informing the development of implementation strategies. Findings from the pilot studies will ultimately inform future research focused on promoting and sustaining youth-derived vaccination service delivery strategies in Nigeria.

## Introduction

Hepatitis B infection (HBV) is of public health significance as it gives rise to substantial morbidity and mortality. According to the World Health Organization, approximately 254 million people had hepatitis B in 2022, including 12% who were children [1]. In 2022, there were 1.2 million new infections globally, with over 6000 new cases each day [1]. Africa bears a disproportionate burden of HBV, with more than 90 million people living with hepatitis B and C in the African Region, accounting for 26% of the global total [2]. In 2019, Nigeria had an HBV prevalence of 8.1% among adults aged 15–64 years [3]. HBV is a leading cause of hepatocellular carcinoma (HCC) and is responsible for an annual incidence of 749,000 new cases of HCC and 692,000 mortalities [4]. HBV vaccine initiated within 24 hours of birth, followed by three additional doses (typically at 6, 10, and 14 weeks after birth), provides durable immunity, substantially decreasing the risk of HBV and the incidence of liver cancer. Hepatitis B vaccines are highly effective in preventing HBV infection. Overall, hepatitis B vaccination produces sero-protection in 98% of healthy term infants. Vaccine response among infants does not vary considerably by maternal HBsAg status or HBIG administration. As a postexposure immunoprophylaxis measure for infants born to an HBsAg-positive birth parent, hepatitis B vaccine without HBIG is 75% effective at preventing perinatal HBV transmission, but when combined with HBIG, the effectiveness is 94% [5]. Nevertheless, only 15 of the 54 African countries currently incorporate the hepatitis B birth dose (HepB-BD) vaccine into their routine vaccination schedules [6]. HepB-BD is included in the Expanded Program of Immunization in Nigeria, which ensures that the vaccine is available to newborns at no cost [7]. However, only half of all newborns are vaccinated against HBV, and the vaccine coverage within 24 hours of birth is 52% [8]. The failure to engage end-users (caregivers of

newborns and expectant mothers) in planning and implementation is a major driver of unsuccessful interventions addressing low vaccination coverage [5].

Countries in the Global South (LMICs) often adopt intervention strategies from the Global North (HICs) without considering local contexts. Modifying these strategies for local relevance is better than wholesale adoption [9]. Many projects in the global South have failed or faced problems because the primary beneficiaries were not involved in initiating, designing, and delivering the project [9]. Community engagement interventions can be sustainable in resource-limited African countries by leveraging local structures and resources. This strategy reduces costs and fosters community ownership. Involving stakeholders, such as community leaders, in designing and implementing these interventions is essential to ensure effectiveness and sustainability [10]. Recently, interest has grown in harnessing youth (15–29) [11] to create solutions for public health issues affecting them and their communities [12,13]. While community ownership is important, emphasizing the agency of youth highlights why they are an ideal target audience. The rationale for youth-led intervention in Vaccination Uptake in Nigeria is that young people represent a significant portion of the population; 45.46% were between 15 and 35 years old in a total population of about 209.6 million in 2021 [14]. Young people are enthusiastic, bring creativity and innovation to outreach, bridge generational gaps, and are future leaders in research. Their ability to mobilize peers, influence social norms, and engage with digital and grassroots platforms positions them uniquely to drive community transformation.

The Participatory Action Research (PAR) and Mega-Designathon approach with youth engagement to address vaccination uptake-related issues is a novel tool ensuring that health interventions are socially and culturally appropriate. By engaging youth in communities, PAR builds trust, empowers participants, creates learning opportunities for the youth, and implements effective, sustainable interventions embraced by the community. This approach strengthens community capacity, promotes leadership, and fosters a collaborative environment for improving health outcomes [15–17]. The PAR approach is informed by an equitable implementation science mindset, which draws explicit attention to the assets, values, and needs of the community in implementing evidence-based interventions with the community rather than just for the community. The Mega-Designathon approach is a time-based, three-stage participatory activity informed by design thinking, intensive collaborative teamwork, and follow-up activities for implementation research. It includes crowdsourcing, intensive collaboration, and an Innovation Bootcamp [18].

This paper details a five-week NIH-funded For Communities by Communities (4CBYC) Innovation Bootcamp within the C3RISE (Cancer Control Center for Research on Implementation Science and Equity) project.

Despite being freely available, delivery and uptake of the hepatitis B birth dose (HepB-BD) vaccine in Nigeria remain suboptimal. To address this gap, we developed a structured Innovation Bootcamp model to train youth in hepatitis B research and implementation science, including but not limited to the epidemiology, burden, prevention, and public health impact, as well as development of implementation research skills.

This study aims to describe our approach and offer insights into how youth-led, crowdsourced strategies can identify barriers and facilitators to HepB-BD uptake and support sustained vaccine delivery in resource-limited settings.

By outlining our journey from generating ideas to developing implementation strategies to be piloted in the field, we aim to offer important insights for researchers keen on implementing Innovation Bootcamps to broaden health service access tailored to communities.

## Materials and methods

### Innovation bootcamp steps

The bootcamp was held in Lagos, Nigeria, from June 10 to July 21, 2024, with the goal of increasing the uptake of HepB-BD vaccination among newborns in Nigeria. The bootcamp was a five-week training program for young people to learn about research principles, entrepreneurship, and program management skills. The bootcamp brought together various communities of interest in a mutually supportive environment to enhance proficiency for youth participants in

research, including ethics training, capacity building in implementation delivery, storytelling, group learning, business model development, and field research. The bootcamp followed a 4-day Designathon with Nigerian youth to generate innovative strategies to increase HepB-BD vaccination uptake in Nigeria.

## Ethics statement

Ethical approval was sought and obtained from the Institutional Review Board of the Nigerian Institute of Medical Research (NIMR #IRB/23/066), ~~a~~ the National ~~e~~Ethics ~~b~~Body authorized to oversee biomedical research involving human participants. The research team ensured that the study was conducted in full compliance with NIMR IRB requirements, as well as international ethical guidelines such as the Declaration of Helsinki. In preparation for fieldwork, both the research team and participants obtained up-to-date certification in human subjects research.

The participants gave written informed consent to participate in the study. Before the commencement of bootcamp activities, trained study staff explained the purpose of the study, the procedures involved, potential risks and benefits, and the voluntary nature of participation. The information was provided both verbally and in writing to ensure understanding. Participants were given sufficient time to ask questions and make an informed decision. The respondents were assured of strict confidentiality regarding all information obtained throughout the study period.

Inclusivity in global research: Additional information regarding the ethical, cultural, and scientific considerations specific to inclusivity in global research is included in the Supporting Information (S1 Checklist).

**Step 1: Identification and recruitment of teams.** Using the Participatory Action Research framework, the project began with an open call contest from February 1 to March 17, 2024, inviting interested participants to generate innovative ways to promote HepB-BD vaccination uptake in Nigeria by responding to an open call contest was advertised through flyers, social media, in-person informational sessions, and emails. Applicants were to submit their response to "*How might we promote birth-dose hepatitis B vaccination for newborns in Nigeria*?" as teams of at least two persons per team.

The research team received 351/362 submissions electronically (WhatsApp, email, and the our 4CBYC website) and 11/362 in person. The entries were evaluated based on relevance, novelty, feasibility, and their potential to promote equity and fairness. All entries were scanned for AI (Artificial Intelligence) usage, and submissions with > 10% of AI usage were automatically disqualified (we used ZeroChatGPT to detect AI-generated content). The top 40 entries were assessed by external judges who narrowed applicants' submissions to the top 10 finalists. The top 10 finalist teams, after signing an informed written consent, were invited to the Designathon, held from April 3rd to 6th, 2024, at the Nigerian Institute of Medical Research (NIMR) in Lagos, Nigeria. During the event, they were mentored by experts in public health implementation science and members of the research team on co-designing and refining their solutions to enhance the uptake and awareness of HepB-BD vaccination among Nigerian communities. At the end of this event the ten teams pitched their implementation ideas to the judges and the five top teams moved to the next stage, being the bootcamp. The stages of the Designathon are shown in Fig 1.

From the Designathon, each team presented a prototype of their HepB-BD vaccination promotion strategy. They addressed the low uptake of the HepB-BD vaccine among Nigerian newborns, outlined the components of their promotional strategy, described team roles and responsibilities, provided a plan for pilot field implementation, and shared projections for implementation outcomes. The five teams that made it past the Designathon and qualified to move to the next stage which was the bootcamp, completed a pre-bootcamp application form that included demographics (e.g., age, place of residence, highest level of education, occupation) and a project proposal (e.g., innovation name and components of the HepB-BD vaccination promotion strategy). The teams ranged in size from two to five individuals, and one team facilitator was assigned to each team to mentor and support them in achieving the boot camp deliverables as needed. The team facilitators were key stakeholders in public health research, communication, entrepreneurship, social innovation, design thinking, youth engagement, social media management, product development, and marketing.

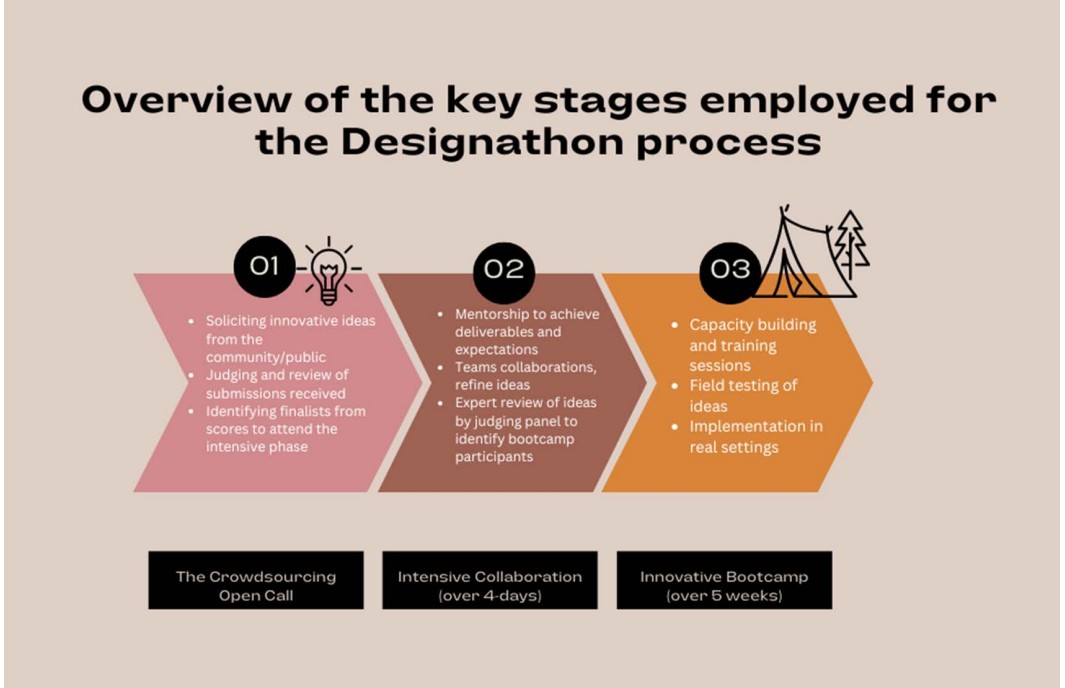

**Fig 1. Overview of the Designathon stages.** This figure illustrates the three main phases of the Designathon approach used to develop community-driven strategies: (01) crowdsourcing an open call to solicit innovative ideas from the public, (02) intensive collaboration involving mentorship and expert review, and (03) implementation through a capacity-building bootcamp with real-world field testing.

**Step 2: Organization of the pre-boot camp activities.** Preparation for the bootcamp began six months before the event and was guided by a modified lean startup journey [19], offering an alternative to traditional academic approaches focusing on translating public health knowledge into practical, sustainable, and scalable solutions. It favors experimentation over elaborate planning and iterative design over traditional "big design up front" development. The process included: (1) formation, (2) validation, and (3) growth. Formation involved the teams further developing their innovation prototypes from the Designathon. The validation phase involved the teams utilizing the "Build, Measure, Learn" feedback loop that took them through the journey of building a Minimum Viable Product (MVP) using ideas generated from the Designathon. Lastly, growth involved working with the teams to identify ways to expand their ideas before the bootcamp. In addition, the research team and facilitators worked with the teams to identify the strengths, weaknesses, opportunities, and threats (SWOT analysis) of the teams' proposed solutions from the Designathon. Other pre-bootcamp activities included recruiting mentors for the instructional component of the bootcamp, recruiting judges for the project evaluations, and event planning logistics (e.g., acquisition of bootcamp venue, organizing transportation for participants, meals, and refreshments, scheduling field trips, and creating the schedule for the bootcamp). The timing of the bootcamp was carefully selected to accommodate the participants' schedules, given that most were university students. Food, transportation, and accommodation were provided for all bootcamp participants.

**Step 3: Implementation of bootcamp activities.** The bootcamp consisted of three main components: (1) virtual instructional seminars and workshops, (2) community assessment field research, and (3) iterative concept development supported by mentors and facilitators. The instructional seminars and workshops laid the foundation for participants to engage in research and transform their ideas from the Designathon into HepB-BD vaccine promotion strategies that could be implemented and tested for feasibility and preliminary efficacy within community settings. Workshops were

conducted by subject-matter experts with experience working with young people. The speakers included entrepreneurs, researchers, communication specialists, youth ambassadors, and community leaders. At the end of the boot camp, teams were expected to deliver four main items: a Practice brief, a completed PLAN (People, Learn, Adapt, Nurture) worksheet, a detailed Persona page, and a Pitch (the 4Ps). The last two days of the bootcamp (July 20–21, 2024) were dedicated to pitching and judging. The judges comprised of community leaders, professionals in the field of research, public health physicians, vaccine advocates, and youth ambassadors. Overall, the teams were encouraged to establish contacts with stakeholders who are integral to the pilot implementation of the HepB-BD vaccination promotion strategies in their communities, complete an innovation plan and study protocol, and present a minimum viable product representing their HepB-BD vaccination promotion innovation at the time of the presentation. The presentation of prizes was performed immediately after the pitch. The Bootcamp training schedule is shown in Table 1.

The pitch involved a team presentation addressing the problem, the solution components of the HepB-BD vaccine promotion strategy, team information and responsibilities, a plan for pilot implementation, and implementation outcome projections, cost model, pilot implementation budget, performance metrics, and sustainability plans. The ranking of the teams was based on the average scores by the judges (60) and average innovation plan scores (40) from the research team. The teams' ideas and presentations were evaluated according to the four criteria, each accounting for 15 points of the total score:

1) Desirability: Does the innovation appeal to the target populations? Does it address the challenges (e.g., low cost, accessibility, acceptability)?

2) Impact: Will the innovation strategy significantly support and champion mothers in vaccinating their newborns? When available, will it reach all or most newborns in Nigeria?

3) Feasibility: Is the innovation feasible/easy to implement? Are the resources available to execute it?

4) Teamwork: How effective are the participants in working as a team, sharing responsibilities, communicating within, and providing mutual support, with effective problem-solving and time-management skills?

Seed funding (Prize money) was provided to the teams immediately after the pitch presentation (HEPCRUSADER, HEXA, M-CHILD, HEROMOMS and HEPRECIATE) towards the pilot of their proposed strategies: NGN 1,000,000 (≈ 700 USD) for first place, NGN 750,000 (≈ 500 USD) for second place, NGN 500,000 (≈ 350 USD) for third place, and NGN 250,000

**Table 1. Bootcamp Training Schedule.**

| VIRTUAL TRAINING June 10 – June 15 | FIELD WORK June 17 – June 21 | BOOTCAMP June 24 – July 21 |
|---|---|---|
| Context: Setting the stage | Field assessment: Sample 16–20 | Community Readiness Assessment Reflection |
| Climate: Understanding community readiness for change | Target:<br>Primary Population<br>Community leaders<br>Influential community members<br>Caregivers and other members of the community<br>Healthcare workers | Research 101 |
| Connection: Fostering community engagement for change | Leadership in the community | Thematic connection |
| Pre-survey | Community Climate | Interactive workshops |
| Research Ethics training | Community knowledge of the issue | Storytelling |
| Field SWOT analysis training | Community Resources | Deliverables: Practice Brief, PLAN worksheet, Persona page |
| Budgeting | Community knowledge of efforts | Pitch, Prize presentation<br>Study Proposal |

(≈ 150 USD) for the fourth and fifth place positions each. All but one out of five (Team HEPRECIATE did not submit a proposal) submitted study proposals that were reviewed and approved by the research team. Although the Innovation Bootcamp was designed to build research capacity among youth, one team (HEPRECIATE) was represented by an older participant (aged 53) due to the lead member's initial difficulty in recruiting a youth collaborator. Her temporary involvement ensured the team's participation during the co-creation, however the team exited the project at the end of the bootcamp.

**Step 4: Post-bootcamp assessment.** A post-evaluation was conducted using a self-administered questionnaire with open-ended questions to assess participants' overall experience with the bootcamp. Participants were recruited to complete the survey during the closing ceremony and while exiting the bootcamp venue the following day. Informed consent was obtained from all participants prior to data collection. The respondents were assured of strict confidentiality regarding all information obtained throughout the study.

From September 18–20, 2024, the four teams participated in a virtual refresher course focused on designing and conducting a pilot study. The research team delivered online lectures, covering topics such as implementation plans, community engagement, data management, research ethics, and conflict resolution. Each team developed a protocol to implement its pilot study, which obtained NIMR Institutional Review Board approval.

**Step 5: Data collection and analysis.** Sociodemographic characteristics, including age, sex, religious affiliation, highest education level attained, and participants' employment status, were obtained before the start of the training. Pre- and post-survey data were collected on changes in knowledge of HBV and research skills, using a structured questionnaire with a 5-point Likert scale (See supplementary file S2 Table) adapted from our previous projects. Specifically, we utilized 12-item (12 questions) 5-point Likert scale questions to assess changes in knowledge of HBV with responses ranging from 1 = Not at all knowledgeable to 5 = Extremely knowledgeable. To assess research skills, we used a 22-item (22 questions) 5-point Likert scale with responses ranging from 1 = not at all skilled to 5 = very skilled. We used the default items that had been developed for prior projects. The survey was then tailored by our Nigerian team to ensure that the HBV-focused components were easily interpretable. Qualitative data about bootcamp participants' preferences and needs were collected pre-training using a semi-structured, open-ended questions, such as: "what specific content would you like to covered during the bootcamp training?" and "what support would you need to have successful training at the bootcamp?". Qualitative data collection proceeded until saturation, defined as the point at which additional interviews yielded no novel insights.

Data analysis was conducted using RStudio version 4.4.2. Descriptive statistics, such as frequencies and proportions, were calculated for socio-demographic variables. For age, both means and medians were reported. We summed the scores across rows for each participant for the different knowledge and research questions to obtain knowledge and research skills scores. Knowledge scores ranged from 12 to 60, while research skills scores fell between 22 and 110, with higher scores reflecting greater improvement. A comparative analysis was conducted using the pre- and post-intervention data to evaluate participants' changes in knowledge and research skills. The Wilcoxon Signed-Rank Test (Exact) was used to compare pre- and post-training data, and results were reported as mean and median differences at 5% level of significance. The qualitative data from the participants' preferences were manually organized into themes and applied to the bootcamp activities.

Moreover, the thematic analysis framework was used to analyze qualitative data from the community needs assessment in a step-by-step process, identifying and synthesizing emerging themes from the pre-intervention survey developed by the teams using Clarke and Braun's (2013) Six-Step Data Analysis Process [20] as shown in Fig 2. The initial coding was conducted by two members of the research team, and themes were iteratively refined through regular discussions and consensus-building. While formal inter-coder reliability measures were not calculated, the process involved collaborative checking and resolution of discrepancies to enhance credibility.

Emerging themes from the implementation strategies proposed by the four teams were categorized utilizing the concepts of the PEN-3 cultural model for the pilot studies. Table 2 shows some of the teams' quotes.

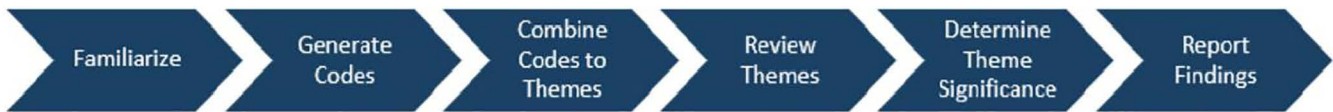

**Fig 2. Clarke and Braun's Six-Step Thematic Analysis Process.** This figure illustrates the six phases of thematic analysis as outlined by Clarke and Braun: (1) Familiarization with the data, (2) Generating initial codes, (3) Combining codes to themes, (4) Reviewing themes, (5) Determining themes' significance, and (6) Reporting findings.

**Table 2. Teams' quotes.**

| TEAM | QUOTE | PEN-3 DOMAIN |
|---|---|---|
| HEXA | "Patients have easy accessibility to vaccines due to close proximity of health facilities" | Enabler Positive |
| HEPCRUSADERS | "By communicating in Yoruba the predominant local language spoken by participants in our location, we ensured clarity, built trust, and eliminated language barriers" | Enabler Existential |
| HEROMOMS | "The participants were initially resistant or hesitant because of the lack of trust in our intervention" | Perception Negative |
| M-CHILD | "We were able to win the trust of the participants by engaging community members, such as leaders, TBAs, and elders" | Nurturers Positive |

## Conceptual framework

PEN-3 cultural model, an equity-centered framework that focuses on Perceptions (beliefs), Enablers (resources), and Nurturers (peers, family, community) within a cultural context that can be used as assets to improve health outcomes, was used by the teams to implement the proposed strategies in the community. PEN-3 ensures interventions are not only evidence-based but also equity-driven, responding to the unique contexts and lived experiences of specific communities; it moves beyond a one-size-fits-all approach, situates health within cultural and social realities, which enhances both its clinical applicability and practical relevance for developing effective and sustainable community-based interventions. PEN-3 model highlights factors within a setting that are positive, existential (inherent to the community, without harmful health consequences), or negative and may influence key stakeholders' and respondents' participation in interventions. The framework comprises three dimensions. The first dimension of the framework, cultural identity, defines the target audience (person, extended family, and neighborhood). The second dimension of the PEN-3 model consists of relationships and expectations, factors like perceptions, enablers, and nurturers. While perceptions represent the knowledge, attitudes, values, and beliefs that play a significant role in the motivation of individuals, enablers are social, system, or structural influences that enhance health beliefs and practices. Nurturers are the role of family and peers in making changes. The third dimension is cultural empowerment. Here, beliefs and customs about health outcomes are identified as positive, negative, or existential [21]. The PEN-3 model is illustrated in Fig 3.

The three conceptual frameworks used in this study are illustrated in Fig 4.

## Results

Five teams, comprising 15 individuals with 2–4 participants per team, completed the bootcamp. As a result, our teams were from the South-West Region and one from the South-East of Nigeria. The mean age was 25.5 years with 14 participants aged 20–28, and one 53-year-old; there were 14 females and one male. Most participants had a secondary school (high school) certificate (53.3%) or a bachelor's degree (40%). Two-thirds of the participants were unemployed at the time of the bootcamp, and 60% had never participated in a bootcamp before the current one. The socio-demographics are shown in Table 3.

PLOS Global Public Health

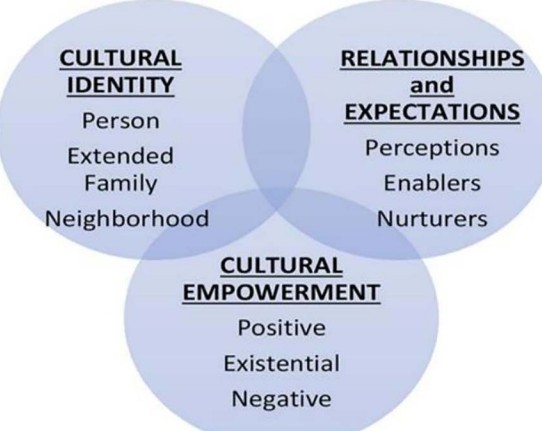

**Fig 3. The PEN-3 cultural model consists of three interrelated domains, which can be used separately or in combination: (1) Cultural Identity (Person, Extended family, Neighborhood); (2) Relationships and Expectations (Perceptions, Enablers, Nurturers); and (3) Cultural Empowerment (Positive, Existential, Negative).** This framework emphasizes the centrality of culture in health behavior and intervention design.

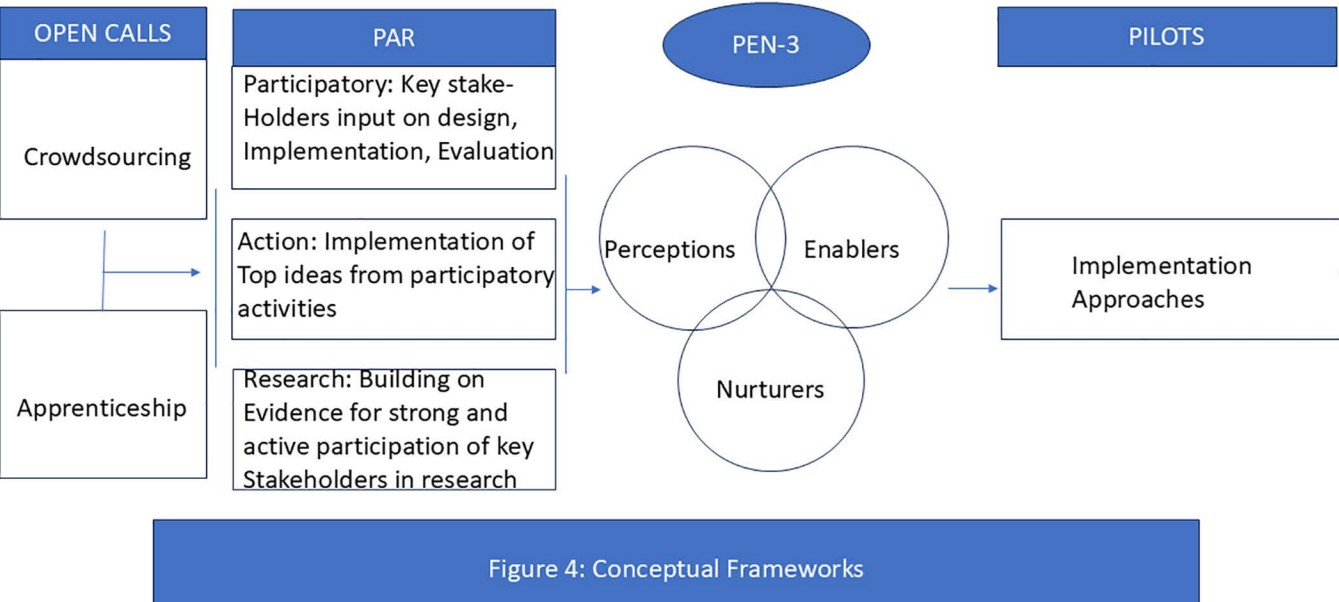

**Fig 4. Conceptual framework integrating cultural constructs of PEN-3 model with participatory research strategies—Crowdsourcing, Designa-thon (Apprenticeship), and PAR—to co-create, implement, and evaluate interventions aimed at increasing uptake of the hepatitis B birth dose vaccination.**

The pre- and post-bootcamp evaluations of participants' responses from the five teams were compared using the Wilcoxon Signed-Rank test, and the results indicated that the specific indicators improved after the bootcamp, which we report here. The median (IQR) score for participants' research skills at the pretest was 57.0 (48.0 – 64.5), compared to 74.0 (70.0 – 81.5) at the post-test ($p < 0.001$). Similarly, the median (IQR) score for HBV-related knowledge at the pretest was 42.0 (37.0 – 44.5), compared to 47.0 (43.0 – 52.5) ($p = 0.010$) at the post-test. According to the survey,

**Table 3. Socio-demographic statistics of bootcamp participants.**

| Variable | Overall (N = 15) |
|---|---|
| **Age** | |
| Mean (SD) | 25.5 (7.89) |
| Median [Min, Max] | 23.0 [20.0, 53.0] |
| **Sex** | |
| Female | 14 (93.3%) |
| Male | 1 (6.7%) |
| **Religion** | |
| Christian | 13 (86.7%) |
| Muslim | 2 (13.3%) |
| **Education** | |
| Bachelor's Degree | 6 (40.0%) |
| Post-Graduate Degree | 1 (6.7%) |
| Secondary School | 8 (53.3%) |
| **Employment** | |
| No | 10 (66.7%) |
| Yes | 5 (33.3%) |
| **Have you participated in an innovation boot-camp before?** | |
| No | 9 (60.0%) |
| Yes | 6 (40.0%) |

participants' skills in conducting research (methods, proposal writing, completing SWOT analysis, engaging stakeholders, ethical considerations, and conducting field pilot studies) and knowledge about hepatitis B (HBV, HepB vaccination, HBV complications) improved significantly compared to the pre-bootcamp evaluation, with increases of 29.82% ($p < 0.001$) and 11.90% ($p = 0.010$), respectively. Descriptive statistics of research skills and HBV-related knowledge are shown in Table 4.

The results from the analysis of the participants' preferences on the content of the training at the bootcamp included immunization history and latest vaccine development, vaccine policies, developing and refining ideas, field research,

**Table 4. Descriptive Statistics of Research Skills and HBV-related Knowledge.**

| Variable | Pre (N = 15) | Post (N = 15) | p-value |
|---|---|---|---|
| Research skills scores | | | <.001 |
| Mean (SD) | 56.1 (11.3) | 76.5 (12.4) | |
| Median [Min, Max] | 57.0 (38.0,74.0) | 74.0 (49.0,101) | |
| 25th Percentile | 48.0 | 70.0 | |
| 75th Percentile | 64.5 | 81.5 | |
| IQR | 16.5 | 11.5 | |
| HBV-related knowledge scores | | | 0.010 |
| Mean (SD) | 39.7 (8.13) | 48.3 (6.95) | |
| Median [Min, Max] | 42.0 (22.0,49.0) | 47.0 (38.0,60.0) | |
| 25th Percentile | 37.0 | 43.0 | |
| 75th Percentile | 44.5 | 52.5 | |
| IQR | 7.50 | 9.50 | |

feasibility of implementation, health promotion and education strategies, data collection, and analysis. The detailed descriptive statistics are found in the supplementary S1 Table.

**Proposed HBV-BD vaccination promotion strategies**

The thematic analysis framework analyzed qualitative data in a step-by-step process to identify and synthesize emerging themes [20]. The key themes emerging from the qualitative analysis of the pre-intervention survey are as follows:

1. Knowledge of the people living in the community about the importance of HepB-BD vaccination (caregivers, mothers, their families, pregnant women).

2. Awareness and knowledge about HBV and its vaccine among traditional birth attendants (TBA), healthcare workers (HCW), educators, and community leaders.

3. The facilitators and barriers that influence the uptake of the HepB-BD vaccination.

4. The effectiveness, acceptability, and feasibility of the health intervention in increasing the uptake of the HepB-BD vaccination.

5. Beliefs, perceptions, and attitudes regarding HepB-BD vaccine.

The proposed strategies addressing the key themes were diverse among the teams, such as training community health workers and traditional birth attendants, educating and sensitizing women in the community, creating a referral system, dialogue with community leaders, and involving families in decision-making, among other strategies, are shown in Fig 5.

This figure presents the community-driven strategies to improve hepatitis B birth dose vaccine uptake developed by four youth-led teams, HEPCRUSADER, HEROMOMS, HEXA, and M-CHILD during the innovation bootcamp. Each team proposed culturally informed, community-based interventions aimed at improving linkage between pregnant women, mothers, traditional birth attendants (TBAs), and formal health systems, increasing public awareness, and strengthening vaccination adherence through education, referral systems, and digital tools. Detailed strategies with reference to the PEN-3 Model are found in the supplement file S2 Table.

The PEN-3 cultural model emphasizes the relationships and expectations domain, which culturally empowers the community to harness its positive assets. These include the safety of vaccines for newborns and their role in preventing illnesses, the availability of free and accessible vaccines, and the support mothers receive from family, peers, and healthcare workers when vaccinating their newborns. Existential perceptions, enablers, and nurturers are unique to the community and deeply embedded in cultural and societal norms, such as the preferred place of giving birth, the language of communication, spirituality and religion, respect for elders' opinions, and beliefs that hospitals are costly. Negative beliefs lead people to think that foreign medical interventions are harmful, that newborns are too frail to receive any vaccination, that vaccines are expensive, that hospital staff are unfriendly, that hospitals are overcrowded, and that immunization schedules are inconvenient. The items to be addressed by the proposed strategies in the bootcamp participants' communities are categorized into the PEN-3 model domains shown in Table 5.

## Discussion

This study demonstrated how a youth-led bootcamp generated community-centered strategies to promote HepB-BD vaccination uptake in Nigeria. Drawing on the PEN-3 cultural model and participatory action frameworks, the bootcamp supported the development of tailored interventions that recognized the complexity of cultural, structural, and informational barriers to timely newborn vaccination. All strategies developed by youth participants highlighted the importance of healthcare facility-based and community-based births for Nigerian newborns. This reflects that the teams comprehended the needs of the target populations in Nigeria. The results align with previous studies, indicating that involving the audience in

**Fig 5. Implementation strategies proposed by the teams.**

**Table 5. The Items addressed in proposed strategies by the teams categorized in the PEN-3 domains.**

| | | Cultural Empowerment | | |
| | DOMAIN | POSITIVE | EXISTENTIAL | NEGATIVE |
|---|---|---|---|---|
| Relationships & Expectations | PERCEPTIONS | • Vaccination is safe for newborns<br>• Vaccines can protect babies from illness | • Traditional birth attendants' care<br>• Hospital births are more expensive<br>• Fear of the unknown | • Foreign interventions are harmful<br>• Believe in traditional/alternative medicine<br>• Newborns are too frail to be vaccinated<br>• Vaccines cost money |
| | ENABLERS | • Healthcare workers' support.<br>• Vaccine accessibility<br>• Vaccines are free<br>• Home visits<br>• Vaccination promotion programs<br>• Incentives and subsidies<br>• Birth registration and vaccination cards<br>• Vaccine policy enforcement | • Language<br>• Respect for elders and community leaders<br>• Cultural events | • Healthcare workers' attitudes.<br>• Crowded healthcare facilities.<br>• Immunization schedules<br>• Births out-of-healthcare facilities are safe |
| | NURTURERS | • Family and peer support of vaccination<br>• Vaccine advocates<br>• Town meetings | • Religion and spirituality | • Misleading information about vaccines from prominent people in the community |

design ensures culturally relevant and logistically feasible outputs that meet community needs [22,23]. Additionally, some proposed innovations included a linkage-to-care element, which is often seen as a barrier to prompt vaccination [24]. Overall, to enhance community awareness and uptake of the HepB-BD vaccine, the teams recommended training and capacity building for traditional birth attendants, educating community health workers, conducting community outreach, hosting workshops, running awareness campaigns, and fostering partnerships between TBA and primary healthcare facilities. One team also suggested an innovative chatbot tool to remind mothers to vaccinate their newborns, expecting that this method would be user-friendly and accessible to Nigerian mothers. Another team proposed implementing a referral system from TBAs to immunization services at primary health centers. The proposed strategies by the four teams converged around three major themes:

### 1. Community engagement and education

Community engagement and education are recurring themes in the proposed teams' strategies. HEPCRUSADER emphasized mobilizing and educating community gatekeepers as HepB-BD vaccine ambassadors to conduct workshops in community health centers and other public sites to reach health workers, pregnant women, and family members. The goal was to educate them on the need for HepB-BD vaccination to prevent liver cancer and dispel any existing misconceptions. Educational materials for these engagements would also be developed. HEROMOMS developed culturally tailored educational materials, including visual aids for TBAs to use during counseling sessions with pregnant women in local languages. Educational materials, such as flyers, would be developed to serve as visual aids for the already trained TBAs during counseling sessions with pregnant women in their local communities' languages. HEXA proposed community dialogues, town hall meetings, and sensitization events while partnering with the Medical Women's Association of Nigeria (MWAN) for advocacy and training. In addition, collaboration with the Medical Women's Association of Nigeria (MWAN), Osun State Chapter will train TBAs, develop resource materials, provide funding, and advocate for policies. Similarly, M-CHILD planned health talks in maternity homes and community events, promoting awareness using the slogan "One Life, One Liver." These strategies reflect the role of tailored, consistent education and community-driven awareness efforts in overcoming vaccine hesitancy and misinformation.

### 2. Training of traditional birth attendants

All four teams recognized TBAs as influential stakeholders for improving newborn vaccination rates. Training TBAs to understand HBV transmission, vaccine benefits, and counseling techniques was prioritized. HEPCRUSADER, HEROMOMS, and HEXA proposed structured workshops to empower TBAs with knowledge and practical tools for community education. M-CHILD integrated TBA engagement with digital innovations, notably using "Ayanfe," a chatbot designed to remind mothers about vaccine schedules. Strengthening the capacity of TBAs is crucial because they often serve as the first and most trusted point of care for pregnant women in Nigerian communities.

### 3. Structured referral systems

Another primary focus was building formal referral pathways between TBAs and primary healthcare centers. HEROMOMS and HEXA proposed linking every TBA home with a specific health facility, with referral cards or tracking systems to ensure newborns are vaccinated promptly. M-CHILD planned to involve local vaccination officers directly in maternity homes to provide on-site HepB-BD vaccination within the first 24 hours. These referral systems aim to bridge gaps between informal birth settings and formal health services, improving continuity of care and ensuring timely immunization.

## Comparison of proposed strategies to scientific literature

Community-based participatory research has gained ground since the 20th century when designing solutions to ensure they fit into the context of the communities for which they are being designed [25]. However, the participatory research

approach is not just a list of strategies for community outreaches, but more of a "systematic effort to infuse community participation in decision-making, local theories of etiology and change, and community practices into the research" [26]. The efforts used for this participatory research were employed in this study by having the interventions designed by young people who belong to the community. Most of the strategies proposed by the teams are found in various literature and have been used in several participatory research studies. One of the teams proposed the use of a chatbot described in the literature, which has shown its feasibility [27]. Three of the four teams have proposed to develop educational materials in collaboration with the TBAs and other key stakeholders in the intervention community as part of their implementation strategy. This has also been utilized by the Community Health Improvement in Milwaukee's Children Project (CHIMC) and fits into one of the key principles of community participatory research; co-learning ensures that input from both the researchers and members of the communities will be used to develop the educational material [28,29]. HEXA intends to leverage the influence of MWAN to push for increased funding and policies that support the integration of TBAs in vaccination programs. This strategy of using community participation to push for policy has been explored by Lucero et al in 2018 [30].

### Potential benefits of youth-driven solutions

Youth-driven research is suited for a larger share of the population, as Nigeria has a large youth population. Also, sustainability becomes more feasible as an increasing number of youths are interested in participating in community projects to enhance their personal and professional development. The use of modern technology can be best understood and developed by the youth, such as the Ayanfe Chatbot by M-CHILD, the tracking and referral system.

### Challenges in implementation

While the proposed interventions by the teams in the Designathon and bootcamp offer promising approaches to improve HepB-BD vaccination uptake among newborns, various challenges across policy, facility, and community levels may make their implementation difficult [31]. Sociocultural factors such as religious practices and beliefs, misconceptions about vaccines, bad experiences with previous vaccination interventions, and the dynamics of decision-making in Nigerian households, where male heads of household heavily influence critical decisions regarding their children's health, may make communities more reluctant to accept the interventions at the community level [26,28,32]. As a limited-resource country, Nigeria faces significant barriers, including shortages of trained personnel, erratic vaccine supplies, unreliable cold chain systems, and long distances between families and healthcare facilities [29,33,34]. Policy and governance issues further complicate implementation, as existing HBV-BD vaccination policies are poorly enforced and inadequately documented [30,35]. A lack of sustained funding and innovation to culturally tailor interventions discourage the long-term implementation of many HBV vaccination initiatives [31,36].

Despite these challenges, discussed during the training, participants rated the bootcamp experience positively. They valued the opportunity to develop research plans, strengthen communication skills, and engage communities through field visits. Beyond enhancing their entrepreneurial and research competencies, participants valued the collaborative and supportive environment fostered by the bootcamp. This participatory approach enhanced technical skills and real-world application, preparing participants to work closely with communities in designing and implementing public health interventions.

Our study had some limitations. Although the pre- and post-improvements in knowledge and skills are statistically significant, the differences observed could be affected by variability or chance, as shown by the overlapping IQRs and the small sample size. It is also important to note that no adjustments were made for multiple comparisons in this study, which may influence the interpretation of the results. Also, the interpretation of pre-post improvements may not be generalizable, given the absence of comparator group, long-term follow-up, and small sample size. The number of finalists was determined pragmatically based on available resources, event logistics, and the capacity to provide meaningful mentorship and support during the bootcamp. The judges selected, to the best of their ability, the most innovative, relevant, feasible, and equitable strategies. Another limitation was that none of the participants originated from the Northen Nigeria; however,

the open call was open to all Nigerian youth and the research team did not have control over the origin of the applicants. To reduce potential selection bias, the contest was promoted both online and offline through various channels, such as webinars, live media sessions, advertisements, radio and television interviews, flyers, banners, and interviews at Nigeria's university campuses and religious centers. This method aimed to reach a diverse audience across different regions, educational levels, and socio-economic backgrounds, including those without reliable internet or social media access. Although some degree of self-selection is unavoidable in open-call models, this inclusive strategy was intended to expand reach and minimize selection bias. The project had only one in-person submission location (Lagos, South-West Nigeria), which may have also contributed to the distribution of entries.

Our findings hold implications for both research and practice. The bootcamp approach extends beyond the conventional idea-generation phase of other participatory methodologies by emphasizing the development and possible execution of intervention strategies. This model integrates elements of capacity-building and field research training programs with the dynamic, collaborative nature of Designathons [16]. From a research standpoint, this study illustrates how the innovation bootcamp model can provide a structured platform for engaging young people, equipping them with skills to design and implement health interventions. The model provides a structured platform for engaging young people and leveraging local knowledge to foster public health innovation. The insights from this study can also guide future interventions and policies, ensuring they are culturally informed, tailored to community needs, and leverage existing community assets and competencies, essential factors for community empowerment and intervention sustainability. This would also strengthen policies implementation.

## Conclusion

Our bootcamp model was practical and leveraged the youth's resourcefulness, capabilities, and resilience to generate community-centered and community-sensitive interventions to promote HepB-BD vaccination. The bootcamp led to the development of four community-centered and participatory social innovations to promote the uptake of HepB-BD vaccination among newborns in Nigeria. The four emerging bootcamp teams are piloting their strategies in their selected communities. The timeline for the screening, recruitment, and implementation of the pilot studies is six months. Findings from the pilot studies will ultimately inform additional research focused on increasing the reach, uptake, and long-term sustainability of youth-derived HepB-BD vaccination service delivery strategies in Nigeria, and reinforce the implementation of the existing policy on HepB-BD vaccination.

Our findings add to the growing body of evidence highlighting the importance of active youth involvement in bridging the HepB-BD vaccination gap among newborns in resource-limited settings such as Nigeria [37]. We contribute to the literature by employing a bootcamp model as a framework to deliver hands-on training and engage youth in designing youth-driven vaccination service delivery strategies for Nigerian newborns. This approach underscores the practicality and benefits of leveraging youth skills and ingenuity to develop health interventions. Such initiatives are more likely to gain community acceptance and sustainability, as they are informed by local experiences, priorities, challenges, perspectives, and solutions.[38].

## Supporting information

**S1 Checklist. Inclusivity in global research.**
(DOCX)

**S1 Table. Summary of pre- and post-surveys assessing skills of bootcamp participants.**
(DOCX)

**S2 Table. Implementation Strategies proposed by four teams in relation to the PEN-3 Model.**
(DOCX)

## Acknowledgments

We acknowledge the Nigerian Institute of Medical Research's efforts in supporting the field work, participants' recruitment, and on-site logistics.

## Author contributions

**Conceptualization:** Maria Augustina Afadapa, Olufunto A Olusanya, Peter Kalulu, Nkiruka Obodoechina, Abideen Salako, Joseph Ogbeh, Juliet Iwelunmor.

**Data curation:** Maria Augustina Afadapa, Olufunto A Olusanya, Peter Kalulu, Abideen Salako, Joseph Ogbeh, Folahanmi T Akinsolu, Ucheoma Nwaozuru, Adesola Z Musa, Dan Wu.

**Formal analysis:** Maria Augustina Afadapa, Peter Kalulu, Joseph Ogbeh.

**Funding acquisition:** Joseph D Tucker, Oliver Ezechi, Juliet Iwelunmor.

**Investigation:** Maria Augustina Afadapa, Olufunto A Olusanya, Nkiruka Obodoechina, Abideen Salako, Joseph Ogbeh, Folahanmi T Akinsolu, Suzanne Day.

**Methodology:** Maria Augustina Afadapa, Olufunto A Olusanya, Peter Kalulu, Temitope Ojo, Collins O Airhihenbuwa, Juliet Iwelunmor.

**Project administration:** Maria Augustina Afadapa, Olufunto A Olusanya, Peter Kalulu, Nkiruka Obodoechina, Temitope Ojo, Abideen Salako, Joseph Ogbeh, Folahanmi T Akinsolu, Titilola Gbaja-Biamila, Suzanne Day, Juliet Iwelunmor.

**Resources:** Olufunto A Olusanya, Nkiruka Obodoechina, Sonam J Shah, Temitope Ojo, Abideen Salako, Joseph Ogbeh, Folahanmi T Akinsolu, Ucheoma Nwaozuru, Titilola Gbaja-Biamila, Collins O Airhihenbuwa, Joseph D Tucker, Oliver Ezechi, Juliet Iwelunmor.

**Software:** Peter Kalulu, Adesola Z Musa, Titilola Gbaja-Biamila.

**Supervision:** Maria Augustina Afadapa, Olufunto A Olusanya, Temitope Ojo, Ucheoma Nwaozuru, Joseph D Tucker, Oliver Ezechi, Juliet Iwelunmor.

**Validation:** Maria Augustina Afadapa, Olufunto A Olusanya, Temitope Ojo, Ucheoma Nwaozuru, Suzanne Day, Peyton Thompson, Kristie Foley, Collins O Airhihenbuwa, Joseph D Tucker, Oliver Ezechi, Juliet Iwelunmor.

**Visualization:** Maria Augustina Afadapa, Olufunto A Olusanya, Peter Kalulu, Nkiruka Obodoechina, Sonam J Shah, Ucheoma Nwaozuru, Olaitan H Olayiwola, Idris A Oladosu.

**Writing – original draft:** Maria Augustina Afadapa, Olufunto A Olusanya, Sonam J Shah, Ucheoma Nwaozuru, Olaitan H Olayiwola, Idris A Oladosu.

**Writing – review & editing:** Maria Augustina Afadapa, Olufunto A Olusanya, Peter Kalulu, Nkiruka Obodoechina, Sonam J Shah, Temitope Ojo, Abideen Salako, Joseph Ogbeh, Folahanmi T Akinsolu, Ucheoma Nwaozuru, Adesola Z Musa, Titilola Gbaja-Biamila, Olaitan H Olayiwola, Idris A Oladosu, Suzanne Day, Peyton Thompson, Kristie Foley, Oluwaseun Falade-Nwulia, Dan Wu, Olufunmilayo Lesi, Collins O Airhihenbuwa, Joseph D Tucker, Oliver Ezechi, Juliet Iwelunmor.

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
