## [Decision Letter · Decision Letter 0]

23 Jul 2025

PGPH-D-25-01335

Improving training on hepatitis B research in Nigeria: Findings from an innovation bootcamp to strengthen capacity

Dear Dr. Afadapa,

Thank you for submitting your manuscript to PLOS Global Public Health. After careful consideration, we feel that it has merit but does not fully meet PLOS Global Public Health’s publication criteria as it currently stands. Therefore, we invite you to submit a revised version of the manuscript that addresses the points raised during the review process.

We look forward to receiving your revised manuscript.

Kind regards,

Katia Bruxvoort, PhD

Academic Editor

Journal Requirements:

Reviewers' comments:

Reviewer's Responses to Questions

**Comments to the Author**

1. Does this manuscript meet PLOS Global Public Health’s publication criteria?

Reviewer #1: Partly

Reviewer #2: Yes

Reviewer #3: Partly

2. Has the statistical analysis been performed appropriately and rigorously?

Reviewer #1: No

Reviewer #2: Yes

Reviewer #3: I don't know

3. Have the authors made all data underlying the findings in their manuscript fully available (please refer to the Data Availability Statement at the start of the manuscript PDF file)?

Reviewer #1: Yes

Reviewer #2: Yes

Reviewer #3: Yes

4. Is the manuscript presented in an intelligible fashion and written in standard English?

Reviewer #1: Yes

Reviewer #2: Yes

Reviewer #3: Yes

Reviewer #1: Summary and Significance:

This manuscript describes a youth-centered, innovation-driven bootcamp conducted in Nigeria aimed at improving research capacity and promoting uptake of the hepatitis B birth dose (HepB-BD) vaccine. The intervention was framed using Participatory Action Research (PAR) and the PEN-3 cultural model, and the outcomes include both knowledge/skills gains among participants and the development of implementation strategies for real-world pilot testing.

Significance:

This is a promising and timely contribution to implementation science, particularly in LMIC contexts. The use of participatory frameworks and youth engagement offers a novel capacity-building model. However, the paper in its current form has serious methodological and reporting limitations that must be addressed to meet the standards of empirical rigor and replicability required for publication in PLOS Global Public Health.

Title and Abstract

•The abstract lacks structured headings, which limits clarity. Consider using a structured format (e.g., Background, Methods, Results, Conclusion).

•The study objective is unclear; the abstract shifts between describing the bootcamp and reporting pre-post results. Clarify whether the manuscript primarily evaluates outcomes or describes the intervention design.

•The phrasing of effectiveness is somewhat overstated given the small sample size and lack of a comparator.

•Key terms like "participatory action research," "PEN-3 model," and "designathon" are introduced without context or brief explanation, which may confuse non-specialist readers.

Introduction

•The public health significance of low HepB-BD coverage is clearly articulated with recent data. However, the introduction does not clearly define the research gap. It would help to explicitly state what has not been done before and how this bootcamp fills that gap.

•Several frameworks (PAR, PEN-3, Mega-Designathon) are introduced, but their relationships and rationale are not well synthesized. A visual model or conceptual diagram could help orient readers.

•The study objective, while stated, is buried in the final paragraph. Consider rephrasing it as a concise, standalone aim statement.

Methods

•Study Design: The manuscript claims a "mixed methods design informed by PAR," but the structure lacks coherence. The study more closely resembles a descriptive implementation study with embedded pre-post assessments and thematic analysis. No formal mixed methods integration is presented.

•Sample Selection: Participant recruitment is described in detail, but several concerns arise:

oSelection appears based on submission to an open call, which may introduce self-selection bias.

oThere is no description of how applicants were screened for eligibility or how representativeness was considered.

oNo justification is provided for the final sample size (N=15), and it is unclear whether this was considered sufficient to detect meaningful changes in pre-post metrics.

•Data Collection:

oThe structure and validation of the pre-post knowledge and research skills survey are unclear. Was the tool piloted? What is the scoring rubric? Were questions validated in a similar setting?

oThe 5-point Likert scales and score ranges (e.g., 1 – 45 for knowledge) need explanation. Composite scores should be justified.

oThe qualitative data collection is described in vague terms (“open-ended questionnaires,” “community needs assessment”), with no mention of interview guide development, piloting, or whether data saturation was achieved.

•Analysis:

oQualitative analysis is said to use Braun & Clarke’s six-step process, but key steps such as theme validation or inter-coder reliability are not described. While the six-step thematic analysis process is referenced, the manuscript does not report whether multiple coders were involved, how disagreements were resolved, or whether any measure of inter-coder reliability was assessed - steps that are critical to ensuring trustworthiness of qualitative findings.

oThe PEN-3 model is presented as a post-hoc framework for organizing results, but the rationale for its application during vs. after coding is not explained.

Results

•Participant Demographics: The sample size is very small (N = 15), heavily skewed by gender (93% female) and age (mean age 25.5). The generalizability of findings is extremely limited and should be acknowledged. The tables are misnumbered – jumping from Table 3 to Table 5.

•Quantitative Results:

oThe pre-post improvements in knowledge (+21.6%) and skills (+36.4%) are statistically significant, but raw score ranges and IQRs suggest considerable overlap. Although the authors report statistically significant improvements in knowledge and skills, the overlapping interquartile ranges and small sample size suggest that observed differences may be due to variability or chance. A more cautious interpretation is warranted, particularly in the absence of a comparator group.

oNo adjustment was made for multiple comparisons, which should be acknowledged.

oResults focus on statistical significance, but clinical or practical relevance is not discussed.

•Qualitative Findings:

oThemes are clearly organized and relevant, but there is no illustrative data (e.g., participant quotes) to ground them.

oThe integration of the PEN-3 model is conceptually sound but would benefit from clearer mapping between themes and PEN-3 components. Figure 3 and Table 5 are helpful, but interpretation is superficial.

•Innovation Strategies:

oOnly four out of five teams completed proposals. This missing data point (HEPRECIATE) should be discussed.

Discussion

•The discussion appropriately emphasizes the potential of youth-led, participatory design for improving public health outcomes. However:

oThere is little reflection on study limitations, including sample size, lack of a control group, non-random selection, and limited external validity.

oInterpretation of pre-post improvements may be overstated, given the absence of a comparator arm or long-term follow-up.

oThe conclusion hints at broader implications for implementation science, but these are not well-developed. How scalable or sustainable is this bootcamp model? What policy implications arise?

oThere’s no discussion of potential biases (e.g., social desirability, recall) or confounders. Given the non-random selection of participants and reliance on self-reported measures, results may be subject to selection and social desirability biases. In addition, without a control group, improvements in knowledge and skills cannot be causally attributed to the bootcamp alone.

Reviewer #2: 1. General Assessment

This manuscript presents a valuable and innovative approach to engaging youth in Nigeria to develop contextually relevant strategies aimed at improving uptake of the hepatitis B birth dose (HepB-BD) vaccine. The participatory action research framework, supported by the PEN-3 cultural model, provides a strong theoretical foundation. The study is commendable for its focus on community-based intervention, youth empowerment, and cultural responsiveness.

2. Originality and Relevance

The manuscript introduces an innovation bootcamp model informed by Participatory Action Research (PAR) and Designathon methodologies—rarely described in the Nigerian public health setting and particularly novel in the context of HBV vaccination.

The study is well-timed, addressing a key challenge in global hepatitis elimination efforts—birth dose vaccine uptake in LMICs, where health system barriers and sociocultural factors remain under-addressed. By emphasizing youth leadership, this study adds a new lens to capacity-building frameworks that typically overlook younger demographics.

Strengths:

a. Grounded in implementation science and cultural theory.

b. Strong cross-sectoral collaboration between Nigerian and US-based institutions.

c. Relevant to the journal’s mission of advancing global public health equity.

3. Methodology

The methodology is comprehensive, detailing each phase of the bootcamp from team recruitment, pre-bootcamp preparation, instructional components, and field implementation to evaluation.

Commendable Aspects:

a. Mixed-methods approach enhances data triangulation.

b. Use of the PEN-3 model is contextually apt for Nigeria.

c. Clear and ethical participant recruitment processes.

d. Pre/post knowledge assessments, use of Wilcoxon tests, and thematic qualitative analysis are appropriate.

Areas for Clarification:

a. Sample Size Justification: Only 15 participants completed the bootcamp. Please elaborate on how this small sample impacts generalizability or whether this was a pilot by design.

b. AI Screening: The manuscript mentions screening for “>10% AI use” during submission evaluation. This is commendable but would benefit from detail—e.g., what tool was used and the rationale for the cutoff?

c. Qualitative Data Saturation: Was thematic saturation reached in the needs assessments?

4. Data Presentation and Analysis

Tables and figures are effectively used. The statistical analyses are appropriate, particularly the use of Wilcoxon signed-rank tests for paired comparisons.

Suggestions:

a. Consider visualizing pre- and post-knowledge scores to enhance interpretation.

b. Include a schematic or visual summary of the PEN-3 mapping for each proposed strategy.

5. Ethical Considerations

The manuscript demonstrates adherence to ethical standards:

Ethical approval and informed consent were obtained.

Participant support (e.g., travel, meals) was provided.

Clarifications Needed:

a. Confirm whether pilot proposals underwent independent IRB review.

b. Address any risk of dual publication: the manuscript cites prior 4YBY bootcamps funded by NIH. Are any datasets or textual components overlapping?

6. Interpretation and Discussion

The discussion is thoughtful and well-grounded in theory and practice. The authors compare their findings to relevant global health and implementation science frameworks and acknowledge both strengths and limitations.

Suggestions:

a. The gender balance was highly skewed (93.3% female). Could this have influenced strategy design, especially in patriarchal settings where male decision-makers may dominate?

b. Further discussion on how the model could be institutionalized (e.g., through government health departments or university curricula) would strengthen the policy implications.

7. Conclusion

The manuscript ends with a strong, practical call to action. The authors provide clear next steps and a justification for further investment in youth-led capacity building for immunization and public health innovation.

8. Additional Comments

a. Language & Editing: The text is clear and well-written; minor grammatical improvements may be needed, please proof read the manucript.

b. Supplemental Files: Ensure all referenced figures/tables (e.g., S1–S4) are uploaded and correctly formatted.

Reviewer #3: Thank you for the opportunity to review this study. Overall, this manuscript described an innovative approach to engaging the community, specifically the youth, to generate creative solutions to addressing barriers to hepatitis B birth dose vaccine uptake among newborns. The approach was a 5-week bootcamp that comprised of activities aimed at building research capacity among young people and supporting the implementation of developed strategies. Though this approach was unique, it is unclear if the authors achieved the study aim, specifically if youth implemented develop strategies as this is not described in the manuscript. Please find attached a detailed list of suggested revisions to help strengthen this manuscript.

**Do you want your identity to be public for this peer review?** For information about this choice, including consent withdrawal, please see our Privacy Policy

Reviewer #1: No

Reviewer #2: **Yes: ** Darlington Faijue

Reviewer #3: No

---

## [Editor Report · Decision Letter 1]

20 Oct 2025

Improving training on hepatitis B research in Nigeria: Findings from an innovation bootcamp to strengthen capacity

PGPH-D-25-01335R1

Dear doctor Afadapa,

We are pleased to inform you that your manuscript 'Improving training on hepatitis B research in Nigeria: Findings from an innovation bootcamp to strengthen capacity' has been provisionally accepted for publication in PLOS Global Public Health.

Best regards,

Katia Bruxvoort, PhD

Academic Editor